# SLIM1 Transcription Factor Promotes Sulfate Uptake and Distribution to Shoot, Along with Phytochelatin Accumulation, Under Cadmium Stress in *Arabidopsis thaliana*

**DOI:** 10.3390/plants9020163

**Published:** 2020-01-29

**Authors:** Chisato Yamaguchi, Soudthedlath Khamsalath, Yuki Takimoto, Akiko Suyama, Yuki Mori, Naoko Ohkama-Ohtsu, Akiko Maruyama-Nakashita

**Affiliations:** 1Department of Bioscience and Biotechnology, Graduate School of Bioresource and Bioenvironmental Sciences, Faculty of Agriculture, Kyushu University, 744 Motooka, Nishi-ku, Fukuoka 819-0395, Japan; yamaguchic124@affrc.go.jp (C.Y.); soudthedlath.most@gmail.com (S.K.); aksuyama@nm.beppu-u.ac.jp (A.S.); y-mori@agr.kyushu-u.ac.jp (Y.M.); 2NARO Tohoku Agricultural Research Center, 4 Akahira, Shimo-Kuriyagawa, Morioka 020-0198, Japan; 3Ministry of Science and Technology, Biotechnology and Ecology Institute, Genetic Resources Division, Don Teaw village, KM 14 office, Tha Ngon Road, Xaythany district, Vientiane 01170, Laos; 4Faculty of Bioscience, Fukui Prefectural University, 4-1-1 Kenjojima, Matsuoka, Eiheiji-town, Fukui 910-1195, Japan; YukiTakimoto@gmail.com; 5Department of Food and Fermentation Sciences, Faculty of Food and Nutrition Sciences, Beppu University, 82 Kita-Ishigaki, Beppu, Oita 874-8501, Japan; 6Institute of Agriculture, Tokyo University of Agriculture and Technology, 3-5-8 Saiwai-cho, Fuchu-shi, Tokyo 183-8509, Japan; nohtsu@cc.tuat.ac.jp; 7Institute of Global Innovation Research, Tokyo University of Agriculture and Technology, 3-5-8 Saiwai-cho, Fuchu-shi, Tokyo 183-8509, Japan

**Keywords:** Cadmium stress, *Arabidopsis thaliana*, sulfate transport, SLIM1 transcription factor, phytochelatin

## Abstract

Sulfur (S) assimilation, which is initiated by sulfate uptake, generates cysteine, the substrate for glutathione (GSH) and phytochelatin (PC) synthesis. GSH and PC contribute to cadmium (Cd) detoxification by capturing it for sequestration. Although Cd exposure is known to induce the expression of S-assimilating enzyme genes, including sulfate transporters (*SULTR*s), mechanisms of their transcriptional regulation are not well understood. Transcription factor SLIM1 controls transcriptional changes during S deficiency (−S) in *Arabidopsis thaliana*. We examined the potential involvement of SLIM1 in inducing the S assimilation pathway and PC accumulation. Cd treatment reduced the shoot fresh weight in the *sulfur limitation1* (*slim1*) mutant but not in the parental line (1;2PGN). Cd-induced increases of sulfate uptake and *SULTR1;2* expressions were diminished in the *slim1* mutant, suggesting that SLIM1 is involved in inducing sulfate uptake during Cd exposure. The GSH and PC levels were lower in *slim1* than in the parental line, indicating that SLIM1 was required for increasing PC during Cd treatment. Hence, SLIM1 indirectly contributes to Cd tolerance of plants by inducing −S responses in the cell caused by depleting the GSH pool, which is consumed by enhanced PC synthesis and sequestration to the vacuole.

## 1. Introduction

Cadmium (Cd) is highly toxic but non-essential for living organisms [1,2,3]. It is released from natural and anthropogenic sources into the environment where it accumulates in the soil. Plants growing in Cd-contaminated soil typically absorb the heavy metal via cation transporters located on the root surface and facilitate its transport to aerial tissues [4,5,6,7,8,9]. Phytotoxic effects of Cd include growth inhibition, leaf chlorosis, and nutrient deficiencies [3,9,10]. However, plants have evolved multiple mechanisms to tolerate Cd exposure [10,11,12,13,14].

It has been demonstrated that Cd is chelated by low-molecular, sulfur (S)-containing compounds, such as glutathione (GSH) or phytochelatin (PC), and sequestrated into the vacuole as GSH-Cd or PC-Cd complexes, respectively [5,10,15,16,17]. Thus, plants can diminish the toxicity of Cd by lowering its cytosolic concentration [11,15,16,18]. GSH is a tri-peptide thiol synthesized from cysteine (Cys), glutamate (Glu), and glycine (Gly) by γ-glutamylcysteine synthetase (GSH1) and glutathione synthetase (GSH2). It contributes to Cd detoxification both as a scavenger of reactive oxygen species (ROS) and as a substrate of PC synthesis [13,19]. Phytochelatin synthase (PCS) synthesizes PC by sequentially adding γ-glutamylcysteine to GSH to generate PC oligomers of variable lengths with the general structure (γ-Glu-Cys)*_n_*Gly (*n* = 2–11) [20]. PC levels and PCS activity are highly enhanced upon Cd exposure, whereas the transcript levels of *PCS* genes, *PCS1* and *PCS2* in *Arabidopsis*, remain relatively constant [21,22]. PCS is activated by binding to heavy metals, resulting in the quick initiation of PC synthesis upon heavy metal exposure [23,24]. The importance of GSH and PC for Cd tolerance is also indicated by the observation that certain mutations in either the *PCS1* or *GSH1* gene are associated with the Cd-sensitive phenotypes of the corresponding *cad1* or *cad2* mutants [25,26,27].

Cd exposure-induced synthesis of GSH and PC stimulates S assimilation by Cys, a major substrate for GSH synthesis [28,29,30,31]. Recent studies reported the close relation between sulfur availability and increased plant tolerance to Cd stress [32,33,34,35,36]. S assimilation in plants starts with the uptake of sulfate by the rhizosphere via sulfate transporters (SULTR) [30,31,37]. Sulfate is absorbed by the root cells and translocated to the plastids, where it is reduced to sulfide by several enzymatic reactions, and assimilated into Cys [28,29,30,31]. The SULTR family in *Arabidopsis* contains 12 members, which can be divided into four distinct functional groups [30,31,37]. Two high-affinity group 1 SULTR proteins, SULTR1;1 and SULTR1;2, facilitate the uptake of sulfate into roots [29,30,36,37]. Group 2 SULTR proteins are low-affinity sulfate transporters that contribute to the sulfate transport through vascular tissues [30,31,37,38,39]. Group 3 SULTR proteins reside in the plastid membrane to ensure the sulfate influx into the plastids [40,41,42]. Group 4 SULTR proteins are responsible for the efflux of sulfate from vacuoles [43]. Transcript levels of the genes involved in S assimilation and GSH synthesis, including *SULTR*, *GSH1,* and *GSH2*, are increased in response to Cd stress, resulting in the upregulation of enzymatic activities [22,44,45,46,47,48,49]. Indeed, Cd-induced sulfate uptake mainly depends on the elevated expression of *SULTR1;2* [50,51]. Cd exposure causes increased sulfate distribution to the shoots, along with an increased sulfate concentration in xylem sap and the upregulated expression of the *SULTR* genes involved in root-to-shoot sulfate transport, such as *SULTR2;1* and *SULTR3;5* [38,39,40,51].

There are many similarities between plant responses to Cd exposure and S deficiency (−S); both stimulate sulfate uptake, sulfate translocation to shoots, and sulfate reduction, along with its assimilation into Cys [22,52,53,54,55]. In −S responses, Sulfur LIMitation1 (SLIM1), an ethylene insensitive 3-like (EIL) family transcription factor, plays a central role in inducing the transcriptional responses [53]. SLIM1 activates sulfate acquisition for S assimilation by upregulating *SULTR1;1, SULTR1;2, SULTR4;1,* and *SULTR4;2* in response to −S conditions [53]. However, it is still unclear how much the −S-induced changes in the transcriptome contribute to the Cd-induced responses and the Cd tolerance in plants. In this study, we investigated the involvement of SLIM1 in plant responses to Cd exposure and its significance in Cd tolerance by comparing the Cd-responsive phenotypes of the *slim1* mutant with those of the parental line *P_SULTR1;2_-GFP* (1;2PGN). The analysis demonstrated the significant contribution of SLIM1 to the Cd-responsive induction of sulfate uptake, sulfate distribution to shoots, and PC accumulation.

## 2. Results

### 2.1. The slim1 Mutant Was More Sensitive to Cd and Accumulated Less Cd Than the Parental Line 1;2PGN

To define the contribution of the SLIM1 transcription factor to plant tolerance for Cd, we compared the effects of Cd treatment on growth and Cd accumulation in a *slim1* mutant, *slim1–2*, with those in the parental line, 1;2PGN (Figure 1a). 1;2PGN is a transgenic line harboring a fusion gene construct consisting of the 2,160 bp 5′-upstream region of *SULTR1;2* and the coding region of *GFP*; this line accumulates GFP in response to −S [55].

When plants were grown without Cd, the fresh weights of shoots and roots of *slim1-2* and 1;2PGN plants were similar. Cd exposure did not affect the root fresh weight in *slim1-2* or 1;2PGN. In contrast, the shoot fresh weight of *slim1–2* was decreased by Cd treatment, whereas that of 1;2PGN was not affected (Figure 1a). Cd tolerance rates described as Cd20/Cd0 rate of shoots fresh weight were 100% and 72% in 1;2PGN and *slim1–2,* respectively.

Then Cd levels in roots and shoots of the plants exposed to 20 µM CdCl_2_ were analyzed (Figure 1b). Cd levels in the root tissues were lower in *slim1–2* than in 1;2PGN, whereas those in the shoots did not vary between the genotypes, as observed in the knockout lines of *SULTR1;2* [51]. These data indicated that SLIM1 contributed to Cd tolerance and accumulation in plants.

### 2.2. Cd-Induced Increases of Sulfate Uptake, Sulfate Transport to Shoots, and Sulfate Content in Shoots Were Diminished in slim1–2

Since *slim1-2* was more susceptible to Cd treatment than 1;2PGN, we further compared the effects of Cd treatment on sulfate uptake, transport to shoots, and sulfate accumulation between *slim1-2* and 1;2PGN (Figure 2).

Cd treatment enhanced sulfate uptake and distribution to shoots in 1;2PGN (Figure 2a). The corresponding increases were not significant in *slim1-2*, and the rates of increase were below those in 1;2PGN. Specifically, sulfate uptake was increased 1.6-fold and 1.4-fold in 1;2PGN and *slim1–2*, and sulfate distribution to shoots was increased 6.8-fold and 3.3-fold in 1;2PGN and *slim1–2*, respectively.

Cd exposure had a similar effect on the sulfate levels in 1;2PGN and *slim1–2* (Figure 2b). When plants were grown without Cd, sulfate levels in shoots and roots did not differ between 1;2PGN and *slim1–2*. In response to Cd exposure, sulfate levels in shoots were increased 1.70-fold and 1.57-fold in 1;2PGN and *slim1–2*, respectively, whereas the root sulfate levels were not affected by Cd exposure in 1;2PGN and *slim1–2* (Figure 2b). These results indicated that sulfate uptake and distribution to shoots under Cd treatment were lower in *slim1–2* than in 1;2PGN.

### 2.3. Cd-Inducible Expression of SULTR1;2 Was Moderated in slim1–2

Sulfate uptake from roots is facilitated by SULTR1;1 and SULTR1;2, whose transcript levels are strongly increased by −S in a SLIM1-dependent manner [53]. Although SULTR1;2 was defined as the main contributor to Cd-induced sulfate uptake in *Arabidopsis* [51], the involvement of SLIM1 in this induction process has not been reported. The transcript levels of two low-affinity sulfate transporter genes, *SULTR2;1* and *SULTR3;5*, which are known to be involved in the root-to-shoot sulfate transport [38,39,40], were increased by Cd exposure and −S [22,38,39,40,54,56], but their −S responses did not depend on SLIM1 [53].

To examine whether SLIM1 mediates the increase in sulfate uptake accompanied by transcript accumulation of *SULTR1;1* and *SULTR1;2* or in root-to-shoot transport of sulfate during Cd exposure, transcript levels of *SULTR1;1*, *SULTR1;2*, *SULTR2;1*, and *SULTR3;5* were analyzed in 1;2PGN and *slim1-2* roots (Figure 3).

Cd treatment did not affect the transcript levels of *SULTR1;1* and *SULTR1;2* in 1;2PGN or *slim1-2* roots. However, *SULTR1;2* expression tended to be stimulated by Cd treatment in 1;2PGN roots but not in *slim1–2* roots (Figure 3) with p values of 0.10 and 0.64, respectively, which were calculated with Student’s t-test, comparing the 0 and 20 µM CdCl_2_ treatment groups. These results suggested that SLIM1 is required for inducing *SULTR1;2* transcription in roots during Cd exposure.

SULTR2;1 is suggested to contribute to the increased root-to-shoot sulfate transport during Cd exposure [50], and the co-expression of SULTR3;5 can stimulate the activity of SULTR2;1 [39]. Cd treatment tended to increase *SULTR2;1* expression in 1;2PGN, and the increase was significant in *slim1–2* (Figure 3), probably because the transcript levels of *SULTR2;1* in shoots is decreased by microRNA395 which expression is induced by −S in SLIM1-dependent manner [56]. The *SULTR3;5* transcript level was not affected by Cd treatment or SLIM1 impairment (Figure 3).

### 2.4. Accumulation of GSH and PC Was Diminished in slim1–2

We measured the total content of S, Cys, GSH, and PC in 1;2PGN and *slim1–2*, to determine the contribution of SLIM1 to the increased thiol accumulation during Cd exposure (Figure 4). When plants were grown without Cd, the total S content in *slim1–2* shoots was lower than that in 1;2PGN shoots; however, under Cd exposure, it was increased to the same level of that in 1;2PGN shoots. Cd treatment caused an increase in the root Cys content in *slim1–2*. The GSH content in the shoots was lower in *slim1-2* than in 1;2PGN under both control and Cd-treated conditions, whereas the root GSH content was similar between control and Cd-treated conditions in 1;2PGN and *slim1–2*.

PCs were barely accumulated in shoot and root tissues of 1;2PGN and *slim1*–*2* under control conditions. Upon Cd exposure, PC levels in both root and shoot tissues of 1;2PGN and *slim1–2* were markedly increased. However, the PC3 and PC4 levels in shoots were significantly lower in *slim1–2* than in 1;2PGN. Less PC3 and PC4 in the *slim1-2* shoots, compared with that in the 1;2PGN shoots, suggested the involvement of SLIM1 in the increase of PC accumulation during Cd exposure.

### 2.5. Cd Treatment and−S Additively Increased SULTR1;2 Expression and Sulfate Uptake, which Depended on SLIM1

Cd-induced contributions of SLIM1 to the increase in sulfate uptake and the increase in PC levels suggested that the response to Cd exposure is partly mediated through −S responses mediated by SLIM1. To demonstrate the involvement of −S in the responses to Cd, *SULTR1;2* expression was analyzed in 1;2PGN and two allelic mutants of *SLIM1*, *slim1-1* and *slim1-2,* under Cd exposure combined with −S (Figure 5).

Plants were grown on agar medium supplemented with 1500 or 15 µM sulfate with or without 20 µM CdCl_2_, and the GFP fluorescence derived from a fusion gene construct, *P_SULTR1;2_-GFP*, was visualized and quantified in roots (Figure 5a,b). GFP signals in 1;2PGN roots were increased by reducing the sulfate levels in the medium, whereas those in *slim1–1* and *slim1–2* were not increased as reported previously [53]. Cd treatment increased GFP fluorescence in 1;2PGN roots in the presence of sulfate at any concentration, which also did not occur in the *slim1* mutants.

To assess whether the GFP levels observed in 1;2PGN and *slim1* mutants were associated with the *SULTR1;2* expression and sulfate uptake, we examined the transcript levels of *SULTR1;2*, sulfate uptake, and the plant growth in 1;2PGN and *slim1* mutants under −S combined with Cd treatment (Figure 6a). The effects of Cd treatment and −S on the *SULTR1;2* transcript level and the GFP fluorescence were similar in the 1;2PGN and *slim1* mutants (Figure 5), which was consistent with the results presented in Figure 2.

The Cd treatment effect on sulfate uptake activity was consistent with that on *SULTR1;2* expression (Figure 6b). Under +S, sulfate uptake activity in 1;2PGN was not significant but tended to be increased by Cd treatment. Sulfate uptake activity in 1;2PGN was increased by −S and enhanced by Cd treatment, whereas that in the *slim1* mutants was not increased by −S and/or Cd treatment.

Fresh weights of shoot and root tissues of 1;2PGN were not affected by −S and Cd treatment (Figure 6c). The fresh weights of *slim1* shoots were lower than those of 1:2PGN when plants were grown with CdCl_2_, corroborating our results shown in Figure 1a. Root fresh weights of *slim1–1* and *slim1–2* were lower than those of 1:2PGN when plants were grown under −S, whereas they were similar to those of 1:2PGN when plants were grown with Cd.

### 2.6. SLIM1 Increases PC Levels in Response to Cd Treatment Even in Combination with−S

Because *SULTR1;2* expression and sulfate uptake were additively stimulated by −S and Cd treatment, and these stimulations were diminished in *slim1* mutants, we examined the thiol content in 1;2PGN and *slim1* mutants under −S combined with Cd treatment (Figure 7). The Cys content in 1;2PGN shoots was not affected by Cd treatment but decreased by −S, whereas that in 1;2PGN roots was increased by Cd treatment and not affected by −S. The Cd-induced increase of the Cys content in roots was more extensive under −S than under +S. The Cys content in *slim1-1* shoots was lower than that in 1;2PGN under +S exposed to Cd. Roots of *slim1* mutants accumulated more Cys under −S as compared to that under +S, but the Cys content did not differ between −S and +S when treated with Cd.

The GSH content of 1;2PGN shoots was decreased by −S and Cd treatment, but it was similar under +S and −S when plants were exposed to Cd. In *slim1* mutants, the GSH content in shoots was less than that in 1;2PGN when plants were grown under −S and/or Cd treatment. However, in 1;2PGN roots, the GSH content was not affected by Cd treatment but decreased by −S. The GSH content in *slim1* roots was lower than that in 1;2PGN when treated with Cd under +S, similar to that in shoots. Under −S, *slim1* roots accumulated more GSH than 1:2PGN roots did, but they had similar GSH levels when exposed to Cd.

The increase in PC accumulation by Cd treatment was moderated in *slim1* mutants under both S conditions (Figure 7). PC was not detected in either 1;2PGN or *slim1* mutants without Cd exposure (data not shown). In 1;2PGN, the PC content in roots did not vary between +S and −S during Cd treatment, but the PC3 and PC4 levels in shoots were significantly decreased under −S. Upon Cd exposure, the PC content was much lower in *slim1* mutants than in 1:2PGN under +S or −S in both roots and shoots. These results suggested that SLIM1 contributed to the accumulation of PC under −S when treated with Cd.

## 3. Discussion

It is well documented that Cd induces PC synthesis and accumulation and enhances sulfate uptake and translocation to shoots in plants [3,9,10,12,13,14,15,20,25,26,27,46,47,51,57]. However, although the expression of genes involved in sulfate uptake and translocation was enhanced under both −S and Cd treatment [38,50,58,59,60], the contribution of −S-induced responses to the Cd-induced responses was not examined. Here we demonstrated the contribution of SLIM1, a central transcription factor coordinating −S responses, to the increased sulfate uptake, sulfate distribution to shoots, and PC accumulation in response to Cd treatment based on the comparison between *slim1* mutants and its parental line 1;2PGN.

Sulfate uptake from roots is facilitated by SULTR1;1 and SULTR1;2, whose transcript levels are highly increased by −S in a SLIM1-dependent manner [50,53,58,59,60,61,62,63,64,65]. The Cd-induced increase of sulfate uptake was suppressed in *slim1* mutants (Figure 2a, Figure 6b), and *SULTR1;2* expression tended to be increased in 1;2PGN but not in *slim1* plants under Cd treatment (Figure 3, Figure 5, Figure 6a). These data indicated that SLIM1 contributed to the increased sulfate uptake by stimulating *SULTR1;2* expression under Cd treatment.

Cd treatment increases sulfate distribution to shoots by stimulating root-to-shoot sulfate transport [51], which probably stimulates the S assimilation by transporting more sulfate to shoots, the main site of S assimilation. The sulfate distribution to shoots and the sulfate content in shoots were increased by Cd treatment in both 1;2PGN and *slim1-2*, but the rate of increase was lower in *slim1-2* than that in 1;2PGN (Figure 2), indicating the involvement of SLIM1 in the increased sulfate distribution to shoots under Cd treatment. However, the induction rates of both *SULTR2;1* and *SULTR3;5* transcripts by Cd exposure were similar between 1;2PGN and *slim1-2* (Figure 3), suggesting the existence of other mechanisms to increase sulfate distribution to shoots beside that coordinated by these SULTRs.

In addition, Cd-induced accumulation of PC was reduced in *slim1* plants together with a lower GSH level in *slim1* mutants (Figure 4, Figure 7), indicating a SLIM1 requirement for PC synthesis and accumulation induced by Cd exposure. The PC shortage in *slim1* mutants probably reduced the chelation of Cd, which resulted in the reduction of shoot fresh weight during Cd exposure (Figure 1a, Figure 6c). Inadequate sulfate uptake in *slim1* mutants should affect PC accumulation through the insufficient supply of Cys for PC synthesis, as observed in the disruption lines of *SULTR1;2* and *SULTR1;1* [51,64]. Thus, during Cd exposure in plants, SLIM1 is responsible for the increased uptake of sulfate, sulfate distribution to shoots, and PC accumulation in plants.

Although the involvement of SLIM1 in the regulation of S metabolism during Cd exposure has been demonstrated, increases of sulfate, thiols, and total S levels under Cd treatment were different from the situation of *slim1* mutants grown under −S in which Cys and GSH levels were approximately 50% and 30% of those grown under S-sufficient conditions, respectively (Figure 2, Figure 4, Figure 7, Figure 8) [51,53]. It is plausible that cytosolic GSH and PC pools can be depleted during Cd treatment because of the sequestration of Cd-GSH and Cd-PC into vacuoles (Figure 8) [66,67], which may elevate the S demand despite the increase of the total S and thiol content (Figure 4). This hypothesis is supported by the observations that providing external Cys or GSH suppresses the expression of S-assimilatory genes, including *SULTR1;1* and *SULTR1;2* [55,66,68], and that the expression of *SULTR1;2* is stimulated by the demand for S [50]. How plants recognize the S status has been a long-debated issue. Studies have identified the roles of various proteins and metabolites, *e.g.*, SULTR1;2, a transporter/receptor sensing the sulfate concentration [69,70], *O*-acetyl-L-serine, a precursor of Cys synthesis that is sensed as the −S indicator [71,72,73], and a toll-like receptor that integrates the S demand and the demand for carbon and nitrogen [74]. Our study revealed the role of the cytosolic GSH level as an indicator of the S status sensed by plants.

The *slim1* mutants had a lower Cd tolerance than 1;2PGN despite of the fact that their roots accumulated less Cd than those of 1;2PGN (Figure 1, Figure 6c). Reduced Cd accumulation in roots associated with a less tolerant Cd phenotype was also observed in the disruption lines of *SULTR1;2* [51]. In addition to this, sulfate application increased Cd concentrations in the roots of wheat, mustard, and buckwheat plant [32,35,75]. However, these findings should be considered in relation to the general suggestion that the ability of plants to accumulate higher heavy metal levels in shoots rather than in roots is associated with a higher heavy metal tolerance [8]. Further research should investigate the effect of the decreased sulfate uptake on the uptake and transport of Cd. Examining the Cd transport in *SULTR1;2* disruption lines and *slim1* mutants may clarify why these plants are less tolerant to Cd.

In conclusion, we demonstrated that SLIM1 contributed to the increased sulfate uptake and PC accumulation under Cd exposure. Our results imply that demand-driven −S responses may be induced in Cd-treated plants, which are stimulated by SLIM1. However, further questions about the underlying mechanisms employed by SLIM1 for regulating these transcriptomic changes under both −S and Cd exposure remain to be clarified.

## 4. Materials and Methods

### 4.1. Plant Materials and Growth Conditions

The *slim1–1*, *slim1–2* and the parental *P_SULTR1;2_-GFP* plants (1;2PGN, *Arabidopsis thaliana* c.v. Columbia; [53,55]) were used as the plant materials. 1;2PGN is a transgenic plant carrying a fusion gene construct that expresses GFP under the control of the 2160 bp 5′-upstream region of *SULTR1;2*, which accumulates GFP simultaneously with the increase of *SULTR1;2* expression [55]. The *slim1* mutants were preselected from the mutagenized M_2_ population of 1;2PGN by screening for the phenotype of impaired GFP fluorescence under –S, and were further tested for the inability to induce a broad range of −S responses, including the upregulation of sulfate uptake, glucosinolate catabolism, and the downregulation of glucosinolate synthesis [53].

The plants were vertically grown on mineral nutrient medium [76,77] containing 1500 µM MgSO_4_, 1% sucrose, and 0.8% agarose, at 22 °C under constant illumination (40 µmol m^−2^ s^−1^) in Figure 1, Figure 2, Figure 3, Figure 4. In Figure 5, Figure 6, Figure 7 agar medium was prepared with washed agar which was washed twice with 6 L of deionized water and 2 L of distilled water followed by the vacuum filtration. To adjust the sulfate concentration in the media, 1500 µM and 15 µM MgSO_4_ were added, and the corresponding media were named as S1500 and S15, respectively. In S15, Mg concentration was adjusted to 1500 µM by adding MgCl_2_. For Cd treatment, the plants were grown vertically for 10 days on the media containing 0 or 20 µM CdCl_2_ (cadmium chloride; Nacalai Tesque, Kyoto, Japan).

Shoot and root tissues were separately harvested, rinsed with distilled water, and used for each analysis. Cd tolerance measurements were performed as previously described [78]. The Cd tolerance rate (%) was calculated by dividing the fresh weights of the Cd-treated plants (n = 4) by the fresh weights of the control plants (Figure 1a).

### 4.2. Cd Analysis

Cd levels were analyzed using an atomic absorption photometer, as described previously [47]. Shoots and roots were dried at 70 °C for 5 days, and digested in 1 mL of HNO_3_. The Cd concentration of each digested solution was measured with an atomic absorption photometer (AAnalyst 200; PerkinElmer, Waltham, MA, USA), using a Cd standard solution (Kanto Chemical, Tokyo, Japan) as a reference.

### 4.3. Sulfate Uptake and Translocation Activity

The sulfate uptake rates and sulfate distribution were analyzed using [^35^S] sodium sulfate (American Radiolabeled Chemicals, St. Louis, Missouri, USA) as previously described [40,63,79]. Plants were vertically grown for 10 days on the media supplemented with 0 or 20 µM CdCl_2_. The roots were submerged in nutrient solution containing 1 MBq [^35^S] sodium sulfate and incubated for 60 min. After several times of wash, radioactivity was measured using a liquid scintillation counter LSC-5100 (Hitachi-Aloka, Tokyo, Japan).

### 4.4. Measurements of Sulfate, Thiols (Cys, GSH, and PCs), and Total S Levels

Plant tissues were frozen in liquid nitrogen and ground with Tissue Lyser MM300 (Retsch, Germany), and then extracted with 5 volumes of 10 mM HCl. The resultant mixtures were centrifuged at 4 °C, 13,000 rpm for 15 min. The supernatant was used for the sulfate and thiol analysis, and the precipitate was used for the total S analysis.

The sulfate content was determined by ion chromatography (IC-2001, TOSOH, Tokyo, Japan), as described previously [40,63]. The content of Cys, GSH, and PCs (PC2, PC3, and PC4) was determined using an HPLC-fluorescent detection system after labeling of thiol bases by monobromobimane as described previously [51,80]. In Figure 4, the labeled products were separated as described previously [51]. In Figure 7, the labeled products were separated using an HPLC system (JASCO) with the TSKgel ODS-120T column (150 × 4.6 mm, TOSOH) and detected with a scanning fluorescence detector FP-920 (JASCO), monitoring for fluorescence of thiol-bimane adducts at 478 nm under excitation at 390 nm. PC2, PC3, and PC4 (Anaspec, Germany), along with GSH and Cys (Nacalai Tesque), were used as standards. The total S content was determined by inductively coupled plasma mass spectroscopy (ICP-MS, Agilent7700x, Agilent Technologies, Santa Clara, CA, USA) as described previously [63].

### 4.5. Quantitative Real-Time PCR (qRT-PCR)

Total RNA was extracted from shoot and root tissues using Sepasol-RNA I (Nacalai Tesque) and reverse transcription was conducted using the PrimeScript RT Reagent Kit with gDNA Eraser (Takara, Japan). Subsequently, qRT-PCR was conducted using SYBR *Premix Ex Taq* II (Takara) and a Thermal Cycler Dice Real Time System (Takara). Relative mRNA abundance was determined using the delta delta Ct (ΔΔCt) method and *ubiquitin2* (*UBQ2*, accession no. J05508) was used as a constitutive internal control. Gene-specific primers for qRT-PCR were previously described [51].

### 4.6. Imaging and Quantification of GFP Fluorescence

The expression of GFP in whole intact seedlings was visualized by using the image analyzer, an Amersham Typhoon scanner 5, equipped with a 525BP20 filter and a 488-nm laser (GE Healthcare, Chicago, USA). Relative intensities of GFP fluorescence were determined by ImageQuant TL as the average intensities in the corresponding areas of root tissues.

### 4.7. Statistical Analysis

In Figure 1, Figure 2, Figure 3, Figure 4 Student’s *t*-test (two-tailed) was used for pairwise comparisons. The Tukey-Kramer test was used for multivariate comparisons. In Figure 5, Figure 6, Figure 7 significant differences among 1;2PGN plants were analyzed for significance using the Tukey-Kramer test, and the significant differences between 1;2PGN and *slim1* mutants under the same treatment were analyzed for significance using Student’s *t*-test.

## Figures and Tables

**Figure 1 plants-09-00163-f001:**
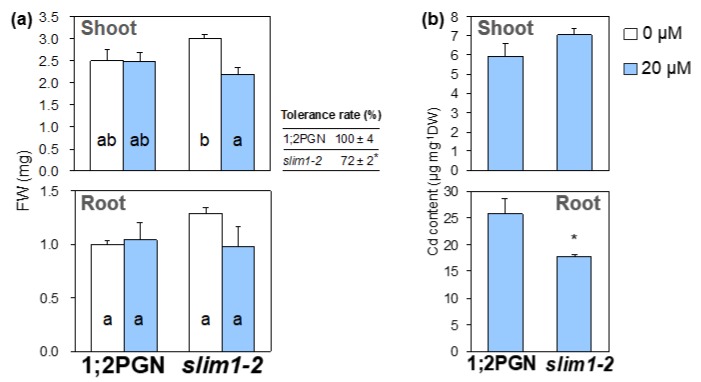
Effects of Cd treatment on plant growth and Cd accumulation in 1;2PGN and *slim1-2*. (**a**) Fresh weight of shoots (upper graph) and roots (lower graph). The Cd tolerance rate (%) was calculated by dividing the shoot fresh weights of the Cd-treated plants by those of the control plants. (**b**) Cd content in shoots (upper graph) and roots (lower graph). Also, 1;2PGN and *slim1-2* were grown for 10 days on MGRL agar medium containing 0 (white bars) or 20 (light blue bars) µM CdCl_2_. The average values are indicated with error bars denoting SEM (*n* = 4). Different letters indicate significant differences between experimental groups (Tukey-Kramer test; *p* < 0.05). Asterisks indicate significant differences between 1;2PGN and *slim1–2* (Student’s *t*-test; * *p* < 0.05).

**Figure 2 plants-09-00163-f002:**
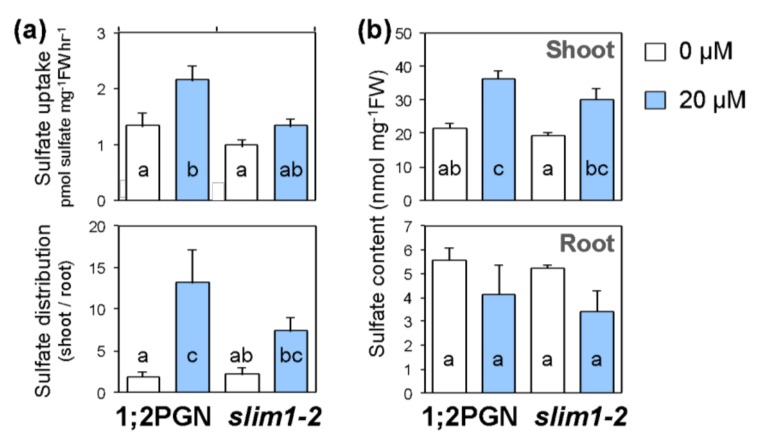
Effects of Cd treatment on sulfate uptake, distribution, and content in 1;2PGN and *slim1–2* plants. (**a**) The absolute values of [^35^S] sulfate uptake rates (upper graph) and sulfate distribution shown as the shoot/root ratio of [^35^S] sulfate accumulations (lower graph). (**b**) Sulfate content in shoots (upper graph) and roots (lower graph) of 1;2PGN and *slim1-2*. Plants were grown for 10 days on MGRL agar medium containing 0 (white bars) or 20 (light blue bars) µM CdCl_2_. Sulfate content was determined by ion chromatography. The average values are indicated with error bars denoting SEM (*n* = 4–7 in a, *n* = 4 in b). Different letters indicate significant differences among experimental groups (Tukey-Kramer test; *p* < 0.05).

**Figure 3 plants-09-00163-f003:**
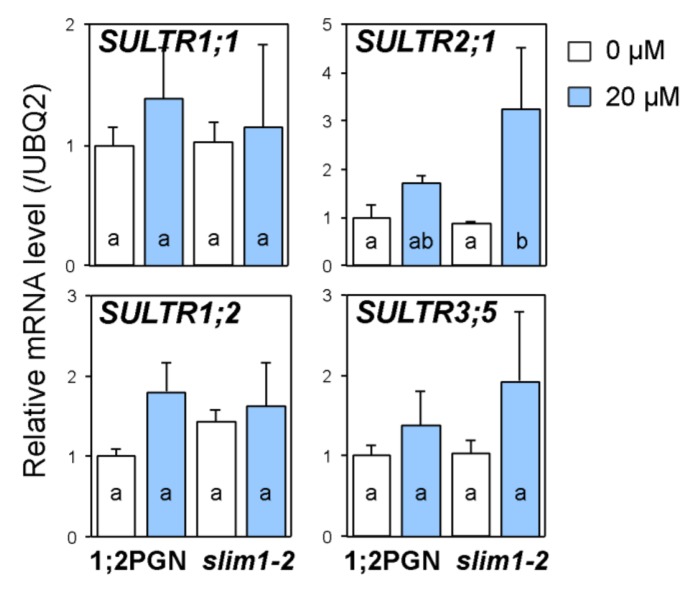
Effects of Cd treatment on transcript levels of *SULTR*s in 1;2PGN and *slim1-2* roots. 1;2PGN and *slim1–2* were grown for 10 days on MGRL agar medium containing 0 (white bars) or 20 (light blue bars) µM CdCl_2_. Transcript levels of *SULTR*s in roots were determined by quantitative real-time PCR (qRT-PCR). The average values are indicated with error bars denoting SEM (*n* = 3–4). Different letters indicate significant differences among experimental groups (Tukey-Kramer test; *p* < 0.05).

**Figure 4 plants-09-00163-f004:**
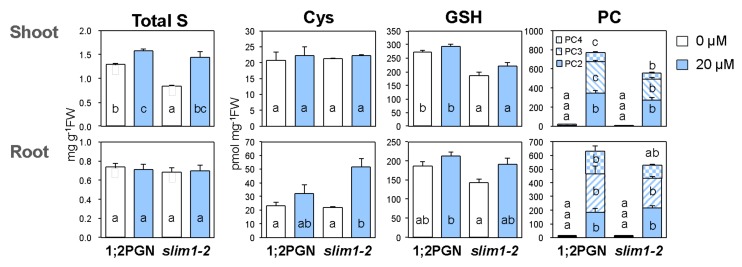
Effects of Cd treatment on the total content of S, Cys, GSH, and PC in 1;2PGN and *slim1–2*. 1;2PGN and *slim1–2* were grown for 10 days on MGRL agar medium containing 0 (white bars) or 20 (light blue bars) µM CdCl_2_. Shoot and root tissues of 1;2PGN and *slim1-2* were used as samples. Total S content was analyzed by ICP-MS. Contents of Cys, GSH, and PCs were analyzed using a HPLC-fluorescence detection system after labeling the thiol bases with monobromobimane. Cys: Cysteine, GSH: Glutathione, PC: Phytochelatin. The average values are indicated with error bars denoting SEM (*n* = 3–4). Different letters indicate significant differences among experimental groups (Tukey-Kramer test; *p* < 0.05).

**Figure 5 plants-09-00163-f005:**
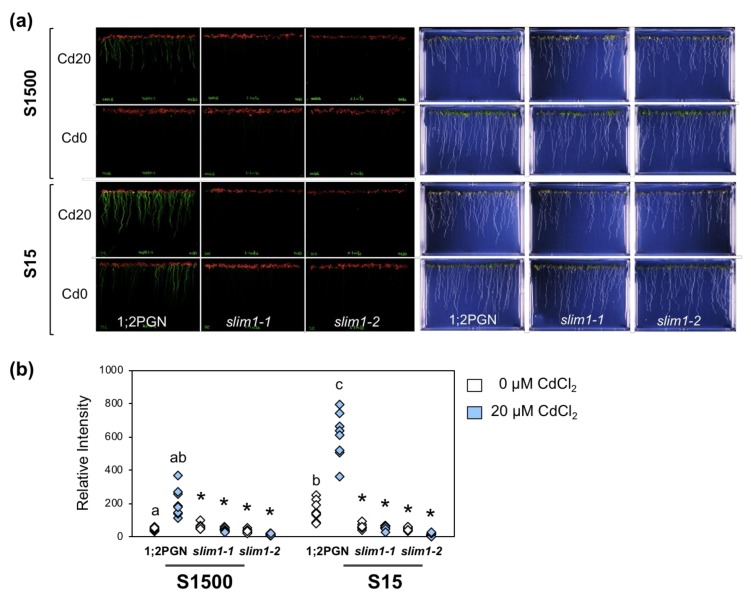
Effects of Cd treatment and −S on GFP and *SULTR1;2* levels in 1;2PGN and *slim1-2* plants. 1;2PGN, *slim1-1* and *slim1-2* seedlings were grown for 10 days on MGRL agar medium supplemented with 1500 or 15 µM sulfate (S1500 or S15, respectively), which contained 0 or 20 µM CdCl_2_. (**a**) GFP fluorescence in *P_SULTR1;2_-GFP* plants was visualized using an image analyzer. Fluorescent images (left panels) and bright field images (right panels) are presented. (**b**) Relative GFP fluorescent intensities in (a) were processed by ImageQuant TL. The fluorescence intensities of plant roots grown on the agar medium containing 0 (white diamonds) or 20 (light blue diamonds) µM CdCl_2_ are displayed (*n* = 6). Different letters indicate significant differences between treatments in 1;2PGN (Tukey–Kramer test; *p* < 0.05). Asterisks indicate significant differences between 1;2PGN and *slim1* mutants (Student’s *t*-test; * *p* < 0.05).

**Figure 6 plants-09-00163-f006:**
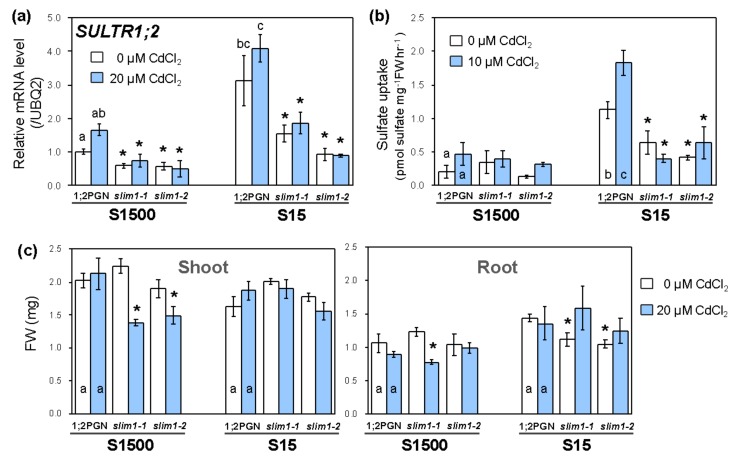
Effects of Cd treatment and −S on *SULTR1;2* levels, sulfate uptake activity, and plant growth in 1;2PGN and *slim1* plants. 1;2PGN and *slim1* seedlings were grown for 10 days on MGRL agar medium supplemented with 1500 or 15 µM sulfate (S1500 or S15, respectively), which contained 0 (white bars) or 10/20 (light blue bars) µM CdCl_2_. (**a**) Transcript levels of *SULTR1;2* in roots were determined by qRT-PCR. The average values are indicated with error bars denoting SEM (*n* = 2–4). Different letters indicate significant differences between treatments in 1;2PGN (Tukey–Kramer test; *p* < 0.05). (**b**) The absolute values of [^35^S] sulfate uptake rates are presented as averages with error bars denoting SEM (*n* = 5–6). Different letters indicate significant differences between treatments in 1;2PGN (Tukey–Kramer test; *p* < 0.05). Asterisks indicate significant differences between 1;2PGN and *slim1* mutants (Student’s *t*-test; * *p* < 0.05). (**c**) Fresh weight of shoots (left graph) and roots (right graph) per plant are presented as averages with error bars denoting SEM (*n* = 3–4). Different letters indicate significant differences between treatments in 1;2PGN (Tukey–Kramer test; *p* < 0.05). Asterisks indicate significant differences among 1;2PGN and *slim1* mutants in the same treatment (Student’s *t*-test; * *p* < 0.05).

**Figure 7 plants-09-00163-f007:**
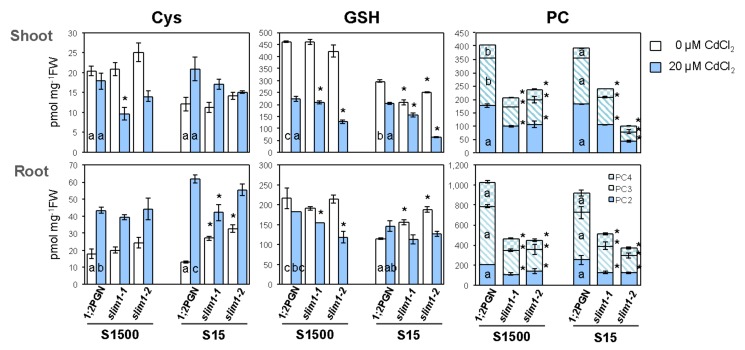
Effects of Cd treatment and −S on the content of Cys, GSH and PCs in 1;2PGN and *slim1* plants. 1;2PGN, *slim1–1,* and *slim1–2* seedlings were grown for 10 days on MGRL agar medium supplemented with 1500 or 15 µM sulfate (S1500 and S15) and containing 0 (white bars) or 20 (light blue bars) µM CdCl_2_. Content of Cys, GSH, and PCs was analyzed by HPLC-fluorescent detection after labeling the thiol bases with monobromobimane. Cys: Cysteine, GSH: Glutathione, PC: Phytochelatin. The average values are indicated with error bars denoting SEM (*n* = 3–4). Different letters indicate significant differences between treatments in 1;2PGN (Tukey–Kramer test; *p* < 0.05). Asterisks indicate significant differences among 1;2PGN and *slim1* mutants using the same treatment (Student’s *t*-test; **p* < 0.05).

**Figure 8 plants-09-00163-f008:**
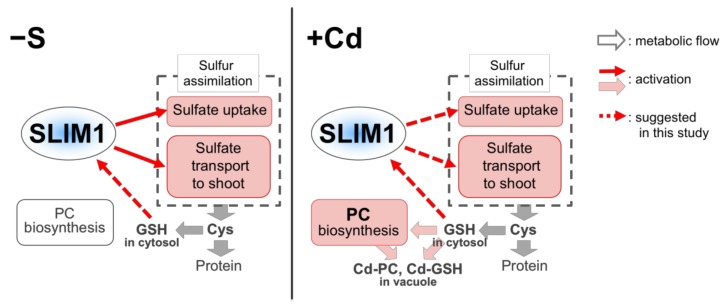
The role of SLIM1 in the regulation of S assimilation under −S and Cd treatment. Left panel: Plants increase sulfate uptake and its translocation to shoots in response to −S. SLIM1 stimulates the expression of genes involved in S assimilation, including *SULTR1;2*. Right panel: Cd treatment increases sulfate uptake, distribution to shoots, and PC accumulation. Shortage of cytosolic GSH can occur because of either sulfate deficiency (left panel) or the enhanced PC synthesis and the compartmentalization of Cd-PC or Cd-GSH to the vacuole (right panel). As a result, SLIM1 is activated in either scenario. Open arrow: metabolic flow; red or pink arrow: activation; dotted arrow: suggested in this study.

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
