# Peer review of "SLIM1 Transcription Factor Promotes Sulfate Uptake and Distribution to Shoot, Along with Phytochelatin Accumulation, Under Cadmium Stress in *Arabidopsis thaliana"

_plants, 2020, doi:10.3390/plants9020163_

Round 1

Reviewer 1 Report

Line 254

Please explain dose Cd

Line 257

Were the Certified reference materials (CRM) used in chemical analysis  of cadmium ?

Author Response

Response to Reviewer 1’s comments

>Line 254: Please explain dose Cd

Thank you for the careful suggestion. We added CdCl2to make the concentration in the media to 20 µM. To avoid the reader’s confusion, we have added “in the media” at the end of the sentence (L360).

>Line 257: Were the Certified reference materials (CRM) used in chemical analysis of cadmium?

Thank you for the careful comments. For Cd analysis, we used a Cd standard solution (Kanto Chemical, Japan) as a reference. The corresponding description were added in Materials and Methods, L366.

Reviewer 2 Report

The authors investigate the involvement of the transcription factor SLIM1 in cadmium tolerance with respect to its role in the Sulfur assimilation pathway. They showed that SLIM1 is contributing to Cd tolerance through regulating the sulfate uptake, glutathione and phytochelatin levels in Arabidopsis. This is an interesting paper, however, has several shortcomings that need to be addressed by the authors. The control plant which is used in this study is not appropriate. Authors should use wild type Arabidopsis instead of 1;2PGN. Why the authors did not compare the mutant phenotype with that of wild type? Furthermore, in most of the studied traits there is no significant difference between 1;2PGN and slim1 which further emphasizes the use of the appropriate control plant in these experiments. Overall, the presented results do not strongly support the conclusion of this study.

Please refer to the below comments:

The introduction lacks the recent research paper regarding Cd tolerance and transport. Some recently published papers can be added to this section.

Figure1. Although the fresh weight in the slim1 mutant is reduced by Cd treatment in comparison with no treatment condition, there is no significant difference between 1;2PGN and slim1 under Cd stress condition. Also, authors need to explain and discuss how slim1 accumulate less Cd in the root but exhibits less tolerance to Cd in compared to the 1;2PGN. Examination of the putative Cd transporter genes may be useful to get a better insight into the observed phenotype.

The authors need to determine the tolerance rate of each plant to show the degree of tolerance/susceptible more clearly. For more details refer to “Environmental and Experimental Botany 166 (2019) 103805”.

Again, in Figure 2 for all the studied traits there is no significant difference between 1;2PGN and slim1 under Cd stress condition. How the authors ascribed the susceptible phenotype of the slim1 to the sulfate uptake and distribution?

Besides, less increase in sulfate uptake in slim1 mutant (figure 1a) is not likely due to the change in the expression level of the genes involved in its uptake since SULTR1;1 and SULTR1;2 did not alter significantly in slim1 in compared to 1;2PGN. It is also interesting that the transcript level of the SULTR1;2 in 1;2PGN is moderately lower than that of slim1?

Lines 281-286: The authors need to give information regarding the PCR condition. Did the authors normalize the transcript level of the examined genes? If so please indicate which method is used.

Use italic for gene names throughout the text.

Author Response

Response to Reviewer 2’s comments

>The authors investigate the involvement of the transcription factor SLIM1in cadmium tolerance with respect to its role in the Sulfur assimilation pathway. They showed that SLIM1 is contributing to Cd tolerance through regulating the sulfate uptake, glutathione and phytochelatin levels in Arabidopsis. This is an interesting paper, however, has several shortcomings that need to be addressed by the authors. The control plant which is used in this study is not appropriate. Authors should use wild type Arabidopsis instead of 1;2PGN. Why the authors did not compare the mutant phenotype with that of wild type?

We appreciateyour critical comments. We agree that the addition of the wild-type, Col, in the analysis would strengthen the conclusion of this study. However, we believe 1;2PGN is an appropriate control for slim1 mutants, because it is the parental line of the slim1mutants and they share the same transgenes. The only difference between 1;2PGN and slim1mutants is the mutation in SLIM1 as the initial slim1mutants were backcrossed 3 times to 1;2PGN.

>Furthermore, in most of the studied traits there is no significant difference between 1;2PGNand slim1which further emphasizes the use of the appropriate control plant in these experiments. Overall, the presented results do not strongly support the conclusion of this study. 

Thank you for the critical comments. We completely agree with your comments and are ashamed of our previous overstatement. In the revised manuscript, we described more carefully and added new data as Figures 5-7, in which we analyzed the effects of Cd under different S conditions. Although the SULTR1;2expression was not statistically different between 1;2PGN and slim1 mutants, the tendency to be increased by Cd in 1;2PGN but not inslim1 mutants were consistent in both Figures 3 and 6a. The same things are applied in sulfate uptake activity analyzed in Figures 2a and 6b, where the activity was increased by Cd in 1;2PGN but not inslim1 mutants in +S and -S conditions. We hope the revision can be acceptable for your requirements.

 >The introduction lacks the recent research paper regarding Cd tolerance and transport. Some recently published papers can be added to this section.

Thank you for the kind advice. We added the recent references in Introduction, L46-47.

>Figure1. Although the fresh weight in the slim1mutant is reduced by Cd treatment in comparison with no treatment condition, there is no significant difference between 1;2PGNand slim1under Cd stress condition.

 We agree with your comments. However, here we focused on the growth retardation caused by Cd treatment and the effects were reproducible between Figures 1 and 6c.

>Also, authors need to explain and discuss how slim1accumulate less Cd in the root but exhibits less tolerance to Cd in compared to the 1;2PGN. Examination of the putative Cd transporter genes may be useful to get a better insight into the observed phenotype.

Thank you for the insightful advices. That is certainly an interesting point considering about the previous studies, thank you for pointing it out. Although we did not analyze the expression of Cd transporter genes, we have added the description in Discussion.

>The authors need to determine the tolerance rate of each plant to show the degree of tolerance/susceptible more clearly. For more details refer to “Environmental and Experimental Botany 166 (2019) 103805”.

Thank you for the helpful comments. We calculated the tolerance rate of each plant according to the way described in “Environmental and Experimental Botany 166 (2019) 103805”. The degree of Cd20/Cd0 in Figures 1-3 demonstrated the Cd response rate of each plant. As a result, we could show the difference in the rate between 1;2PGN and slim1mutants.

>Again, in Figure 2 for all the studied traits there is no significant difference between 1;2PGNand slim1under Cd stress condition. How the authors ascribed the susceptible phenotype of the slim1 to the sulfate uptake and distribution?

Thank you for the critical comments. As mentioned above, we have changed the description to that sulfate uptake and distribution were increased by Cd in 1;2PGN, but not in the slim1 mutants, which demonstrated that SLIM1 was required for the Cd-induced increase of sulfate uptake and distribution.The results were supported by the results in Figures 5 and 6.

>Besides, less increase in sulfate uptake in slim1mutant (figure 1a) is not likely due to the change in the expression level of the genes involved in its uptake since SULTR1;1and SULTR1;2did not alter significantly in slim1 in compared to1;2PGN. It is also interesting that the transcript level of the SULTR1;2 in1;2PGN is moderately lower than that of slim1?

 Thank you for the thoughtful comments. As pointed, we can say only there were the tendency to be increased by Cd treatment and the tendency was reproducible in Figure 6a. Off course we cannot deny the other reasons for the increased uptake of sulfate, however, considered from the previous report indicating SULTR1;2 is the main factor to the increased sulfate uptake under Cd exposure, we think it is the reasonable discussion. To avoid the overstatement, we have moderated the descriptions in the revised manuscript.

 >Lines 281-286: The authors need to give information regarding the PCR condition. Did the authors normalize the transcript level of the examined genes? If so please indicate which method is used.

We appreciate your careful comments. We described the normalization method for RT-PCR in Materials and Methods, L395-397.

>Use italic for gene names throughout the text.

We appreciate your careful comments.We corrected the gene names to italic.

Reviewer 3 Report

The manuscript entitled “SLIM1 transcription factor promotes sulfate uptake and distribution to shoot, along with phytochelatine accumulation, under cadmium stress in Arabidopsis thaliana” by Chisato Yamaguchi, Soudthedlath Khamsalath, Yuki Takimoto, Akiko Suyama, Yuki Mori, Naoko Ohkama-Ohtsu and Akiko Maruyama-Nakashita describes the potential involvement of SLIM1 transcription factor in Cd-induced S assimilation. I found this manuscript interesting, describing important pathway in plant tolerance towards cadmium however some major issues need to be addressed before this manuscript can be recommended for publication in Plants.

1. Title/keywords: should be phytochelatins instead of phytochelatine

2. In general introduction provides comprehensive background but some additional information about SULTR transporters should be added including how many SULTR proteins is present in A. thaliana. Also there is not enough information about SLIM1 transcription factor (what other genes are controlled by this transcription factor).

3. I am not convinced by some of the presented results:

- Fresh mass of shoot of slim1-2 mutant is lower on media containing Cd than on control media but the differences between the fresh mass of the parental line and slim1-2 on Cd media are not statistically significant (fig. 1a)

- Sulfate uptake and distribution from shoot to root is higher for both lines (parental and mutant) on Cd-containing media and there is a tendency that is higher in parental lines but again the differences are not statistically significant (fig 2).

- There is no statistically significant differences in SULTR 1;1, 1;2 3;5 expression. And for SULTR 2;1 the expression is higher on Cd media in mutant line which suggests the SLIM1-independent pathway (fig. 3).

- Total S content is higher in slim1-2 on Cd-media than on control media which suggests SLIM1-independent mechanism of sulfate uptake but again is the same as for control media. Cys content is higher in slim1-2 on Cd-media than on control media which also suggests SLIM1-independent mechanism. The only thing that is lower in slim1-2 mutant on Cd-media in comparison to its parental line is phytochelatins but only in shoots and only PC3 and PC4.

4. Authors did not present results to support some of their conclusions:

“ The Cd-induced increase of sulfate uptake was suppressed in slim1-2 (Figure 2a), and SULTR1;2 expression tended to be enhanced in 1;2PGN but not in slim1-2 under Cd treatment (Figure 3).”

The differences between parental line and slim1-2 line were not statistically significant both for sulfate uptake and SULTR1;2 expression.

“The sulfate distribution to shoots and the sulfate content in shoots were increased by Cd treatment in both 1;2PGN and slim1-2, but the rate of increase was lower in slim1-2 than that in 1;2PGN (Figure 2a), indicating the involvement of SLIM1 in the increased sulfate distribution to shoots under Cd treatment”

The differences between parental line and slim1-2 line were not statistically significant for sulfate distribution.

Lower level of GSH in shoots in slim1-2 mutant in comparison to parental line on Cd-media was observed which probably caused also lower level of PCs which suggest the involvement of SLIM1 in S metabolism also in response to Cd toxicity but in my opinion additional experiments are needed to gain the proper insight into this issue. The final conclusions are not fully supported by the presented results.

5. I am not convinced that slim1-2 mutant is a proper model plant. The parental line is already modified line with insertion of GFP under SLUTR1;2 promoter. It should be established whether changes in response to Cd and -S are visible for this line in comparison to WT therefore WT should also be used as additional control. Authors should consider generation a knock-down or knock-out mutant in SLIM1 gene using WT line. I like the aim of the study and I think that it is important to check whether the Cd-induced response overlaps with -S induced response but some additional experiment are needed.

Author Response

Response to Reviewer 3’s comments

>The manuscript describes the potential involvement of SLIM1 transcription factor in Cd-induced S assimilation. I found this manuscript interesting, describing important pathway in plant tolerance towards cadmium however some major issues need to be addressed before this manuscript can be recommended for publication in Plants.

We appreciateyour critical comments. According to the comments, we have revised the manuscript. In the revised manuscript, we described more carefully about the statistical differences and added new data as Figures 5-7, in which we analyzed the effects of Cd under different S conditions. We hope the revision can be acceptable for your requirements.

>1. Title/keywords: should be phytochelatins instead of phytochelatine

Thank you for your careful comments. We have revised Title and keywords.

>2. In general introduction provides comprehensive background but some additional information about SULTR transporters should be added including how many SULTR proteins is present in A. thaliana. Also there is not enough information about SLIM1 transcription factor (what other genes are controlled by this transcription factor).

Thank you for the careful advices. We have added theinformations about SULTR and SLIM1 in Introduction, L67-72 and L81-84, respectively.

>3. I am not convinced by some of the presented results:

- Fresh mass of shoot of slim1-2 mutant is lower on media containing Cd than on control media but the differences between the fresh mass of the parental line and slim1-2 on Cd media are not statistically significant (fig. 1a)

We agree with your comments. However, here we focused on the growth retardation caused by Cd treatment and the effects were reproducible between Figures 1 and 6c.

> Sulfate uptake and distribution from shoot to root is higher for both lines (parental and mutant) on Cd-containing media and there is a tendency that is higher in parental lines but again the differences are not statistically significant (fig 2).

We agree with your comments. However, here we focused on the plant responses to Cd. There was a significant increase of sulfate uptake in the parental line when it was treated with Cd, but there was no effect on that in the slim1mutants, indicating the response is dependent on SLIM1. The effects were reproducible between Figures 2 and 6b.

> There is no statistically significant differences in SULTR 1;1, 1;2 3;5 expression. And for SULTR 2;1 the expression is higher on Cd media in mutant line which suggests the SLIM1-independent pathway (fig. 3).

Thank you for the thoughtful comments. Under −S, plants induces the expression of microRNA 395 (miR395) in a SLIM1-dependent manner, which results in the degradation of SULTR2;1mRNA in shoots [51, 52], supporting the different induction of SULTR2;1levels in response to Cd in 1;2PGN and slim1shoots is also due to the  SLIM1. We added the corresponding description in Results, L158.

>Total S content is higher in slim1-2 on Cd-media than on control media which suggests SLIM1-independent mechanism of sulfate uptake but again is the same as for control media. Cys content is higher in slim1-2 on Cd-media than on control media which also suggests SLIM1-independent mechanism. The only thing that is lower in slim1-2 mutant on Cd-media in comparison to its parental line is phytochelatins but only in shoots and only PC3 and PC4.

Thank you for the critical comments. We completely agree that PC3 and PC4 levels in shoots are the only difference between 1;2PGN and the slim1mutants in Figure 4. According to the suggestion, we have moderated the description. Actually, slim1mutants can still induce -S responses as it is a single amino acid substitution. We speculate the same thing can be applied for Cd responses. In Figure 7, GSH levels in shoots and PC levels were decreased in slim1under both +S and -S conditions. Although the effects of Cd on thiol accumulation was fluctuated between the Figures, but the decrease of PC is definitely reproducible.

>4. Authors did not present results to support some of their conclusions:

“ The Cd-induced increase of sulfate uptake was suppressed in slim1-2 (Figure 2a), and SULTR1;2 expression tended to be enhanced in 1;2PGN but not in slim1-2 under Cd treatment (Figure 3).”

The differences between parental line and slim1-2 line were not statistically significant both for sulfate uptake and SULTR1;2 expression.

“The sulfate distribution to shoots and the sulfate content in shoots were increased by Cd treatment in both 1;2PGN and slim1-2, but the rate of increase was lower in slim1-2 than that in 1;2PGN (Figure 2a), indicating the involvement of SLIM1 in the increased sulfate distribution to shoots under Cd treatment” The differences between parental line and slim1-2 line were not statistically significant for sulfate distribution.

Thank you for the critical comments. As mentioned above, we are focused on the plant responses to Cd, so that when there is a significant difference between Cd treatment in 1;2PGN but not in slim1mutants, the above description is appropriate, we think.

>Lower level of GSH in shoots in slim1-2 mutant in comparison to parental line on Cd-media was observed which probably caused also lower level of PCs which suggest the involvement of SLIM1 in S metabolism also in response to Cd toxicity but in my opinion additional experiments are needed to gain the proper insight into this issue. The final conclusions are not fully supported by the presented results.

 Thank you for the critical comments. We hope the addition of new data could provide the better insight to the manuscript.

>5. I am not convinced that slim1-2 mutant is a proper model plant. The parental line is already modified line with insertion of GFP under SLUTR1;2 promoter. It should be established whether changes in response to Cd and -S are visible for this line in comparison to WT therefore WT should also be used as additional control. Authors should consider generation a knock-down or knock-out mutant in SLIM1 gene using WT line. I like the aim of the study and I think that it is important to check whether the Cd-induced response overlaps with -S induced response but some additional experiment are needed.

We appreciateyour critical comments. We agree that the addition of the wild-type (WT) in the analysis would strengthen the conclusion of this study. We also agree that if we compare the effects of Cd between knock-out mutant and WT, the differences can be more relevant. However, we still believe slim1 mutants and 1;2PGN can be the model plants in this kind of comparison, as the both plants share the same transgenes and the only difference between 1;2PGN and slim1mutants is the mutation in SLIM1 because of the 3-time backcrossing to 1;2PGN.

         Thank you for pointing the importance to check whether the Cd-induced response overlaps with -S induced response. According to the suggestions, we have added new data as Figures 5 to 7 to further illustrate the overlaps between Cd and -S responses. Our results suggested that the response to Cd exposure is partly mediated through −S responses controlled by SLIM1.

Round 2

Reviewer 2 Report

Please refer to the comments below:

1- Figures (1 and 4) are mixed and not distinguishable.

2- Regarding the lack of the recent research paper in the introduction, the authors only add a review paper which is not the point here.

3- While they claimed that they calculate the tolerance rate according to the suggested manuscript, but I failed to see any update. If they want to calculate the tolerance rate it needs to be added in the M&M section.

4- Gene names still in some cases are not italic. For instance line 144.

5- Please be consistent regarding the qPCR results. If it is normalized to the UBQ level it should be updated in figure 6a.

6- change Real-time quantitative PCR to qPCR or quantitative Real-time PCR.

Author Response

Response to Reviewer 2’s comments

1) Figures (1 and 4) are mixed and not distinguishable.

Thank you for your careful comments. In Figures 1 and 4, the effects of Cd treatment on plant growth and Cd accumulation and the effects of Cd treatment on the total S, Cys, GSH, and PC, were demonstrated. We think they are distinguishable.

We suppose this comment was pointing that Figures 1 and 4 were similar with Figures 6 and 7. Plants were treated partly the same conditions in between Figures 1 and 4 and Figures 6 and 7, respectively. However in Figures 6 and 7, we have added another allele of slim1 mutant and the plants were treated with the combinations of Cd and the different sulfur conditions.

2) Regarding the lack of the recent research paper in the introduction, the authors only add a review paper which is not the point here.

Thank you for your careful comments. We added the recent research papers in Introduction, L64-65, and Discussion, L414-415.

3) While they claimed that they calculate the tolerance rate according to the suggested manuscript, but I failed to see any update. If they want to calculate the tolerance rate it needs to be added in the M&M section.

Thank you for the helpful comments. We calculated the Cd20/Cd0 ratio of each plant according to the way in the suggested reference. The results were presented in Figure 1a and described it in Results, L123-124. The calculation methods were described in Materials and Methods, L481-590.

4) Gene names still in some cases are not italic. For instance line 144.

We appreciate your careful comments. We corrected the gene names to italic. We did not italicized when it meant protein name.

5) Please be consistent regarding the qPCR results. If it is normalized to the UBQ level it should be updated in figure 6a.

We appreciate your careful comments. We corrected the vertical axis label to explain that RNA levels were normalized by UBQ level in Figure 6a.

6) change Real-time quantitative PCR to qPCR or quantitative Real-time PCR.

Thank you for your careful comments. We have revised Real-time quantitative PCR to quantitative Real-time PCR.

Reviewer 3 Report

The manuscript entitled “SLIM1 transcription factor promotes sulfate uptake and distribution to shoot, along with phytochelatin accumulation, under cadmium stress in Arabidopsis thaliana” by Chisato Yamaguchi, Soudthedlath Khamsalath, Yuki Takimoto, Akiko Suyama, Yuki Mori, Naoko Ohkama-Ohtsu and Akiko Maruyama-Nakashita has been significantly improved. I have only a few minor comments:

Additional information has been added to the introduction and now it provides comprehensive background. Page 2 line 70 – should be group instead of Group Because results and discussion are separate authors should not comment the gained results in the result section but all conclusions should be rather presented in the discussion e.g. page 3 lines 116-117, page 5 lines 168-169, page 6 line 203, page 7 lines 230-232, page 9 lines 280-281. Figure 6 and figure 7 – please consider another way to label statistically significant differences since now it is very confusing. Materials and methods – please add some information about sulfate and the concentration of sulfate in MGRL medium should also be added somewhere in the text. What was the concentration of sulfate in plant media used in experiments which results are presented in figures 1-4? General text editing and polishing is required.

Author Response

Response to Reviewer 3’s comments

>Page 2 line 70 – should be group instead of Group

We appreciate your careful comments. We replaced Group to group in L69.

>Because results and discussion are separate authors should not comment the gained results in the result section but all conclusions should be rather presented in the discussion e.g. page 3 lines 116-117, page 5 lines 168-169, page 6 line 203, page 7 lines 230-232, page 9 lines 280-281.

Thank you for your careful comments. As we think these interpretations are necessary to connect the former results to the next experiments, we decided to maintain the sentences. The instructions for author of “Plants” also allow it as written in research manuscript sections, “Results: Provide a concise and precise description of the experimental results, their interpretation as well as the experimental conclusions that can be drawn.”

>Figure 6 and figure 7 – please consider another way to label statistically significant differences since now it is very confusing.

Thank you for your helpful comments. We also think it is confusing that the way to label statistically significant differences with both alphabets and asterisks. Actually, we had tested other ways, but we could not find a good way to show the combinatorial effects of Cd/−S treatment and the differences between 1;2PGN and slim1 mutants together. Now we believe that the current way represents the effects of both environmental factors and genetic factors well.

>Materials and methods – please add some information about sulfate and the concentration of sulfate in MGRL medium should also be added somewhere in the text. What was the concentration of sulfate in plant media used in experiments which results are presented in figures 1-4?

We appreciate your careful comments. We have added the descriptions about the sulfate concentrations in agar media in Materials and Methods, L436-442.

>General text editing and polishing is required. 

Thank you for your advice. The revised manuscript has been proof read by a professional scientific language editing service prior to submission. Please find the certificate attached.
